# Relationship between Macrophages and Tissue Microenvironments in Diabetic Kidneys

**DOI:** 10.3390/biomedicines11071889

**Published:** 2023-07-03

**Authors:** Jiayi Yan, Xueling Li, Ni Liu, John Cijiang He, Yifei Zhong

**Affiliations:** 1Division of Nephrology, Longhua Hospital, Shanghai University of Traditional Chinese Medicine, Shanghai 200032, China; yanjiayi99@163.com (J.Y.); xuelingli@shutcm.edu.cn (X.L.); annie0722@shutcm.edu.cn (N.L.); 2Department of Medicine, Division of Nephrology, Icahn School of Medicine at Mount Sinai, New York, NY 10029, USA

**Keywords:** diabetic nephropathy, macrophages, inflammation, fibrosis, single-cell RNA sequencing, cell–cell interaction, microenvironment, therapeutics

## Abstract

Diabetic nephropathy (DN) is the leading cause of end-stage kidney disease. Increasing evidence has suggested that inflammation is a key microenvironment involved in the development and progression of DN. Studies have confirmed that macrophage accumulation is closely related to the progression to human DN. Macrophage phenotype is highly regulated by the surrounding microenvironment in the diabetic kidneys. M1 and M2 macrophages represent distinct and sometimes coexisting functional phenotypes of the same population, with their roles implicated in pathological changes, such as in inflammation and fibrosis associated with the stage of DN. Recent findings from single-cell RNA sequencing of macrophages in DN further confirmed the heterogeneity and plasticity of the macrophages. In addition, intrinsic renal cells interact with macrophages directly or through changes in the tissue microenvironment. Macrophage depletion, modification of its polarization, and autophagy could be potential new therapies for DN.

## 1. Introduction

Diabetic nephropathy (DN) is one of the main microvascular complications in patients with diabetes mellitus (DM) and is the leading cause of end-stage kidney disease [1]. The pathogenesis of DN is very complex. After Hasegawa et al. first proposed in 1991 that tumor necrosis factor-α (TNF-α) and interleukin-1 (IL-1) were involved in the pathogenesis of DN [2], more and more epidemiological and preclinical findings have since suggested that systemic and local renal inflammation plays a key role in the development and progression of DN. A large number of studies have shown that inflammatory cells, cytokines, chemokines, adhesion molecules, and other inflammatory factors and immune mechanisms are involved in the pathogenesis of DN [3]. For example, clinical studies have suggested that the circulating soluble TNF-α receptors 1 and 2 can be used as important predictive biomarkers for DN progression [4]. Some of these inflammatory factors play a chemotactic role, which promotes the recruitment of immunoinflammatory cells with macrophages as the predominant immune cell type [5] at the site of inflammation, leading to inflammation and fibrosis through the release of inflammatory mediators, reactive oxygen species, and anti-angiogenic factors. Infiltration of immunoinflammatory cells can be observed in the glomeruli and interstitium of renal biopsy samples at all stages of DN [6]. Many preclinical studies have shown that the therapy targeting the innate immune pathway of DN has achieved promising results [7], and several clinical trials of anti-inflammatory therapy have demonstrated a good short-term efficacy [8], thereby confirming an important role of the inflammatory microenvironment in both the diabetic kidneys and in the progression of DN.

Macrophages are the main immune cells involved in the pathogenesis of DN. Analysis of animal models of Type 1 and Type 2 DN have shown that cluster of differentiation (CD)68+ macrophages account for 90% of the total kidney leucocyte infiltrate and can be derived from either monocyte recruitment or through local proliferation [5]. A diabetic microenvironment, including high glucose and advanced glycation end products (AGEs) can promote the expressions of cytokines, chemokines, and adhesion molecules in renal intrinsic cells, which recruit and activate macrophages [9]. Both the activation of resident macrophages and the increased infiltrating macrophages from the circulating monocytes in diabetic kidneys promote renal inflammation in glomeruli and the tubulointerstitium [10], thereby creating an inflammatory microenvironment in diabetic kidneys.

This review described the interactions between macrophages, intrinsic renal cells, and the tissue microenvironment in the diabetic kidney in order to better understand the roles of macrophages in DN. We also summarized the potential therapeutic strategies by targeting macrophages to prevent and treat DN. In addition, we provided several new concepts and perspectives based on recently obtained new findings from single-cell RNA sequencing studies of macrophages in diabetic kidneys.

## 2. Macrophage Polarization in DN

Macrophages are a type of phagocytic immune cell which can be derived from the differentiation of blood monocytes or exist as resident cells in tissues from early embryonic development [11]. The origins and specification of kidney macrophages have been well reviewed [12]. Studies have shown that most kidney macrophages are derived from embryonic progenitors, which initially migrate from the yolk sac and later from the fetal liver into the developing kidney, and then colonize the kidney during its development, and proliferate in situ throughout adulthood. In the kidney, tissue-specific transcriptional regulators induce the differentiation of macrophage progenitors into dedicated kidney macrophages. In addition, hematopoietic stem cells (HSCs) from the aorta–gonad–mesonephros region also enter the fetal liver and contribute to these resident macrophage populations. Some HSCs can migrate into the bone marrow and the spleen, where they are maintained before being released into the bloodstream postnatally as circulating monocytes that can contribute to the tissue-resident macrophage populations. The following four types of macrophages can be typically detected in specific tissues during steady or stimulated states: tissue resident macrophage, monocyte-derived inflammatory macrophage, constitutive tissue macrophages deriving exclusively from monocytes, and monocytes migrating through tissues [11]. These immune cells can adopt a range of phenotypes and functions based on their microenvironments, allowing them to contribute to both homoeostasis and disease [13]. Therefore, macrophages have high plasticity to adapt in different microenvironments.

Many studies have suggested that macrophage accumulation in kidneys correlates strongly with renal function, inflammation, and kidney cell damage. Though recent studies indicated that macrophages have more complicated phenotypes, most studies on DN have only focused on the traditional M1 and M2 macrophages [14]. It has been shown that M1 and M2 play the opposite role in renal inflammation [15]. At the early stage of kidney injury, resident macrophages, which have an intrinsic ability for self-maintenance through proliferation [11], are activated by pathogen-associated molecular patterns (PAMPs), danger-associated molecular patterns (DAMPs) [16], interferon-gamma (IFN-γ), and pro-inflammatory cytokines [14] to differentiate into proinflammatory M1 macrophages, which are formed in response to infection or cellular damage. Simultaneously, circulating monocytes are recruited to the kidney to differentiate into pro-inflammatory M1 macrophages. Therefore, both the activation and infiltration of macrophages in the diseased kidneys are due to changes arising from the kidney microenvironment. M1 macrophages exhibit proinflammatory effects with the high expression of inducible nitric oxide synthase (iNOS). M1 macrophages secrete pro-inflammatory cytokines (TNF-α, IL1b, and IL-6) to produce an inflammatory microenvironment in the kidney and promote tissue damage [14]. Persistent M1 infiltration and its associated inflammation lead to a decreased renal function and eventually fibrosis [17].

M2 macrophages are normally induced by interleukin 4 (IL-4) and interleukin 13 (IL-13) [18], which suppress inflammation and promote wound repair and fibrosis [19]. M2 macrophages exhibit immunomodulatory and repairing effects with a high expression of CD206, CD163, arginase-1 (Arg-1), and mannose receptor (MR). M2 macrophages have anti-inflammatory effects by secreting anti-inflammatory molecules, such as IL-10, but also promotes renal fibrosis [20] at the late stage of DN by secreting profibrotic cytokines, such as transforming growth factor-beta (TGF-β). A study demonstrated that the activation of Wnt/β-catenin signaling contributed to kidney fibrosis by stimulating macrophage M2 polarization [21]. M2 macrophages could be further divided into M2a, M2b, M2c, and M2d subcategories. These macrophages differ in their cell surface markers, secreted cytokines, and biological functions. However, studies have indicated that the induction routes and regulated biological processes of M2 macrophages are complex interlacing network systems rather than simplistic schema [22]. Therefore, the significance of this sub-classification remains unclear.

The phenotypes of M1 and M2 macrophages have been mostly defined by in vitro studies, and its exact role in the regulation of renal inflammation and fibrosis in vivo remains unclear. Several studies have shown that when inflammation is suppressed, M2 macrophage infiltration does not associate with fibrosis, regardless of the disease duration. However, another study showed that the depletion of inflammatory M1 macrophages does not protect against kidney fibrosis, while the depletion of anti-inflammatory and reparative M2 macrophages can reduce kidney fibrosis [23]. These inconsistent observations may suggest that the timing for macrophage removal may be a crucial component of such strategies to favor renal repair. In addition, the depletion of one type of macrophage may cause a shift from another type of macrophage due to the diseased microenvironments (Figure 1). Therefore, it might be better to target the diseased microenvironment instead of the macrophage subtypes.

## 3. Recent Novel Findings from Single-Cell RNA Sequencing

Recent single-cell RNA sequencing (scRNA-seq) studies have helped us to better define the macrophage subpopulation based on their unique molecular markers [24]. However, the availability of these scRNA-seq studies to analyze the macrophage function in DN are quite limited due to the following reasons: (1) overall, only a few scRNA-seq studies have been published in DN, and (2) most studies were performed using single nuclear RNA sequencing, which is good to preserve the integrity of the tubular cells but is unable to catch the information of the immune cells. In our first study [5], we performed scRNA-seq analysis of isolated glomerular cells from streptozotocin-induced, diabetic *eNOS*-deficient (*eNOS*^−/−^) and control *eNOS*^−/−^ mice. We found that immune cells increased in the glomeruli of diabetic mice and further that most of them were macrophages. Further cluster analysis revealed that they were predominantly M1 macrophages. To capture the gene expression changes in specific macrophage cell subsets in DN we performed a scRNA-seq analysis of CD45-enriched kidney immune cells from type 1 diabetic OVE26 mice at two time points (3 and 7 months, respectively) during the DN progression and performed a detailed analysis of mononuclear phagocytes in the diabetic kidney (Figure 2) [25]. Interestingly, our studies suggested that both the resident and infiltrating macrophage subsets increase in the diabetic kidneys over time, and that both M1 and M2 macrophages increase in the kidney from mice with early DN. The differences between our first study [5] and this study [25] are as follows: (1) we used different animal models of DN, (2) we used different sequencing methods, and (3) most importantly we sequenced glomeruli in the first study, while we sequenced whole kidney cortices in the second study, which contains mostly the tubulointerstitial compartment. Therefore, we believe that in early DN, glomeruli may have predominant M1 macrophages, while both M1 and M2 macrophages increase in the tubulointerstitial compartment.

For this scRNA-seq study of immune cells from the kidney cortices of diabetic OVE26 mice, we performed further analysis of the macrophage polarization states in each subset of macrophages, which showed changes that were consistent with the continuum of the activation and differentiation states, and that their gene expression tended to shift towards undifferentiated phenotypes but with increased M1-like inflammatory phenotypes in the diabetic kidneys compared to the controls. A comparison of several differentially expressed genes in specific macrophage subsets showed consistent findings in human DKD bulk RNAseq by deconvolution analysis [26]. For example, we demonstrated that several subclusters of macrophages change in the diabetic kidney, such as TREM2 (triggered receptor expressed on myeloid cells 2) macrophage subsets. Our study confirms the heterogeneity of the macrophage subsets and a dynamic change in the macrophage phenotypes in the diabetic kidney.

Consistent with our findings, Subramanian A et al. [27] performed scRNA-seq in both mouse and human kidneys with diabetes and obesity. They used two different mouse models: a high-fat diet (HFD) model and a genetic diabetic model (BTBR ob/ob), which is known to develop DN. Interestingly, they also found an increased Trem2^high^ macrophage population in the kidneys of HFD mice that matched with human TREM2^high^ macrophages in obese patients. These data together with our findings suggest a critical role of TREM2 macrophages in DN. However, it is unclear whether TREM2 macrophages have renal protective or damaging effects in DN. In addition, it would be interesting to assess how changes in the kidney microenvironment could affect the phenotypes of macrophages by scRNA-seq.

## 4. Increase of Macrophages in Human DN

The studies of macrophages in human DN are limited to the staining of macrophages and their subpopulations using different markers. These studies suggest that macrophages are the most prevalent infiltrating leucocyte found in human diabetic kidneys, and are associated with the declining renal function observed in patients with DN. Several groups have investigated the activation status and phenotypes of macrophages, which change significantly in the kidney during the progression of DN, likely due to the changes in the kidney microenvironment. Compared to patients without renal abnormalities and from diabetic patients without DN, the number of macrophages increase in both the glomeruli and tubulointerstitium of Type 2 diabetic patients with DN [6], and the infiltrating macrophages are mainly M1 macrophages [28]. Furthermore, what is more interesting is that researchers revealed a 2:1 ratio of M1:M2 cells in both the glomeruli and tubulointerstitium [6]. Moreover, deconvolution analysis of the RNA sequencing data set showed a significant increase in macrophages in the patients with advanced DN compared to those with early DN and the controls [26]. Indeed, staining of M1 and M2 macrophages in the diabetic kidney suggests that both M1 and M2 macrophages increase in DN as compared to the normal controls. However, M1 macrophages were first recruited into the kidney in the early stage of DN while the number of M2 macrophages caught up during the late stage. Therefore, the M1/M2 ratio peaked during the early stage of DN (ratio = 2), and then dropped in the late stage of DN (ratio = 1) [29]. Together, macrophages infiltrating the kidney in early DN have a predominantly M1-type proinflammatory phenotype which promotes kidney injury. These macrophages then locally transform to M2 macrophages, which are the predominant phenotype in the convalescence and repair phases and are thought to be the major players leading to the development of fibrosis later in the disease. Nevertheless, the simple M1/M2 classification of macrophages based on the staining of 1–2 markers is not enough to characterize the complicated functions of macrophages in human DN. In addition, the M1/M2 switch is a dynamic process which is dependent on the kidney microenvironment during the disease progression [30]. Therefore, further studies are required to further dissect this dynamic process using scRNA-seq similar to those we have performed in diabetic mice.

## 5. Interactions between Macrophages and Intrinsic Renal Cells in DN

### 5.1. Glomerular Cells

Glomerular cells include mesangial cells, glomerular endothelial cells (GECs), and podocytes. Our single-cell RNA sequencing studies of glomeruli from DN mice suggest that glomeruli has increased macrophages in early DN with predominant M1 macrophages [5], indicating that macrophages may be involved in glomerular cell injury in early DN. M1, but not alternatively M2 macrophages were found to directly contribute to glomerular injury in DN, which was partially mediated by the increasing podocyte permeability and the eventual impairment of podocyte function. Furthermore, podocytes treated with high-glucose promoted macrophages migrate and accumulate through secreting monocyte chemotactic protein-1 (MCP-1) [31].

Podocytes are highly differentiated epithelial cells attached to the glomerular basement membrane. As the outermost layer of the glomerular filtration barrier, podocytes are an important part of the glomerular filtration barrier and play an important role in maintaining the normal filtration function of the kidney. Other studies have shown that TNF-α released by macrophages under high-glucose conditions can promote the apoptosis of podocytes [32]. It has been shown that Tim-3 induces macrophage activation, which in turn accelerates podocyte damage via the NF- κ B/TNF- α pathway [33]. MiR-21-5p in macrophage-derived extracellular vesicles (EVs) increases ROS production through the targeted inhibition of A20, leading to podocyte damage [34]. It has also been shown that vitamin D [35], FK506 [36], and the overexpression of Sirt6 [37] can reduce podocyte damage through inhibiting the activation of M1 macrophages. These data suggest a close interaction between the macrophages and the podocytes.

In the early stage of DN, glomerular hyperfiltration and hypertrophy were observed, and the number of GECs were increased and associated with enlarged glomerular tufts, suggesting that the mechanism was similar to angiogenesis. Vascular endothelial growth factor (VEGF) was proven to be chemotactic for macrophages, NO was found to negatively regulate VEGF-induced macrophage migration by inhibiting Flt-1 expression, and endothelial nitric oxide synthase (*eNOS*) knockout mice have been well known to be able to develop advanced glomerular lesions resembling human DN [38]. In addition, the vascular endothelium-overexpressed cell adhesion molecules were also present in its surface, such as intercellular adhesion molecule-1 (ICAM-1) [39] and vascular cell adhesion molecule-1 (VCAM-1) [40], which homed circulating macrophages. Recent studies also showed that Exendin-4 (a GLP-1R agonist) can directly act on the GLP-1R on GECs to reduce the expression of ICAM-1 and thereby inhibit macrophage infiltration [41]. Furthermore, it has also been discovered that the accumulation of M1 macrophages upregulates the ROS level in human GECs to promote cell damage [42]. On other hand, injured GECs upregulate the HIF-1α/Notch1 pathway in DN leading to M1 macrophage recruitment, which was reversed by the PPAR-α agonist fenofibrate to improve the GEC function [43]. Due to the critical crosstalk between the macrophages and the GECs, scientists have built a logic-based differential equations model to assess the macrophage-dependent inflammation in GECs during DN progression [44].

### 5.2. Tubular Epithelial Cells

Renal tubular epithelial cells (RTECs) line the renal tubules, and act as not only the target but also as the mediator of macrophage-mediated tubulointerstitial inflammation [45]. RTEC injury releases cytokines and chemokines, which causes the recruitment and activation of macrophages in the diabetic kidney [46], while activated macrophages release more cytokines to cause more tubular cell injury, thereby forming a vicious cycle. MCP-1 [47], as well as osteopontin [48] expressed by RTECs are critical factors which play a key role in the communication between the injured RTECs and the infiltrating macrophages under high-glucose conditions. Toll-like receptor 2 (TLR2), which was implicated in the innate immune response, was proven to be highly expressed in both the glomeruli and the tubulointerstitium and was found to be associated with an increased renal expression of MyD88 and MCP-1, activation of NF-κB, and the infiltration of macrophages [49]. However another study showed that the increased expression of TLR4 but not of TLR2 in the renal tubules of human kidneys with DN correlated with interstitial macrophage infiltration as well as tubulointerstitial inflammation [50]. In addition, macrophages may participate and promote the necroptosis of RTECs under high-glucose conditions, which could be inhibited by the necroptosis inhibitor necrostatin-1 [28]. Previous studies also showed that high glucose can stimulate IL-1β expression in the RTECs to induce the M1 polarization of macrophages [51].

### 5.3. Extracellular Vesicles

Extracellular vesicles (EVs) are a heterogeneous population of membrane-bound vesicles that are considered to be central messengers for intercellular communication. They are released by the majority of cell types and can be classified into two main categories depending on their origin: exosomes, which are released by the endosomal compartment, and ectosomes, which are released through the budding of the plasma membrane. It has been shown that extracellular vesicles (EVs) also mediate communications between damaged intrinsic renal cells and macrophages in the diabetic microenvironment, which provided a new standpoint for the treatment of patients with DN. miR-25-3p in exosomes produced by M2 macrophages protected podocytes against HG-induced injury through the activation of autophagy in podocytes via inhibiting dual-specificity protein phosphatase 1 (DUSP1) expression [52]. miR-93-5p expression was markedly upregulated in lipopolysaccharide (LPS)-induced podocytes, one of the DN models in vitro, and inhibition of miR-93-5p or silencing of TLR4, which was a downstream target of miR-93-5p, reversed the reno-protective effects of miR-93-5p-containing exosomes produced by M2 macrophage on LPS-induced podocyte injury [53].

RTEC-to-macrophage communication forms a feedback loop via EVs to promote RTEC apoptosis, renal inflammation, and tubulointerstitial fibrosis in DN [54]. Studies found that EVs from HSA-treated RTECs can accelerate macrophage glycolysis by stabilizing HIF-1α expression [55]. High levels of miR-19b-3p were found in urinary EVs and were correlated with the severity of tubulointerstitial inflammation in patients with DN [56]. Tail-vein injections of miR-199a-5p, which was found to be increased in urinary EVs from diabetic patients with macroalbuminuria, induced kidney macrophage M1 polarization, and accelerated the progression of DN by targeting the Klotho/TLR4 pathway [57]. Furthermore, miR-7002-5p in EVs derived from high glucose-induced macrophages suppressed the autophagy of RTEC by targeting Atg9b, leading to renal tubular dysfunction and inflammation [58]. A study found that Epsin1 modulated tubulointerstitial inflammation via regulation of exosomal-Dll4 release from the RTECs, which then activated Notch 1 signaling in macrophages under DN conditions [59]. Moreover, macrophages have been identified as the main source of myofibroblasts via macrophage–myofibroblast transition, thereby promoting renal fibrosis [60]. It has also been shown that exosomes from high glucose-treated macrophages promote RTECs to switch to a more pro-fibrosis phenotype via releasing long non-coding RNAs [61]. However, the role of exosomes in mediating intercellular communication requires further investigation.

We summarized the interactions between macrophages and intrinsic renal cells in DN into the table below (Table 1).

## 6. Potential Therapeutics for DN by Targeting Macrophages

### 6.1. Depletion of Macrophages

Macrophage accumulation can lead to the development and progression of DN and therefore the depletion of macrophages could be a potential therapeutic approach.

Many strategies have been used to reduce macrophage infiltration in the kidney to reduce renal injury and inflammation. Systemic irradiation of the kidney, polyclonal anti-macrophage sera, and the micro-encapsulated toxic drug dichloromethylene diphosphate have been used to deplete macrophages in the kidney. However, such systemic macrophage depletion strategies may lead to the alteration of the systemic immune defense system. For DN, it has been shown that macrophage depletion in diabetic CD11b-DTR mice significantly attenuated albuminuria, kidney macrophage recruitment, and glomerular histological changes [31]. However, these studies need to be further confirmed, and in addition, the systemic effects of such an approach need to be considered.

### 6.2. Targeting the Tissue Microenvironment to Modulate Macrophage Infiltration and Function

Recent studies have focused on targeting the kidney tissue microenvironment that leads to macrophage infiltration and activation locally in the kidney [62,63]. For example, blocking chemokines, adhesion molecules, and their receptor expression in renal tubular epithelial cells could reduce macrophage infiltration locally in the kidney. Studies showed that mice genetically deficient in chemokine-CC motif ligand 2 (CCL2) have a reduced capacity to recruit and activate kidney macrophages and are substantially protected from the development of DN [64]. PQ529 (QC/isoQC inhibitor) can cause the degradation of CCL2 and reduce the activation of macrophages, thereby reducing renal inflammation and further kidney damage [65]. Piezo1 can increase macrophage accumulation by upregulating CCL2, the C-C motif chemokine receptor 2 (CCR2) pathway, and the knockout of Piezo1 in mice can inhibit macrophage infiltration, inflammation, and renal fibrosis [66]. In addition, MCP-1 inhibition [67] and CCR2 antagonists have been shown to improve renal injury, albuminuria, fibrosis, and loss in renal function [68,69,70]. Targeting the kidney tissue microenvironment is deemed as a better approach as it can avoid systemic effects.

### 6.3. Drugs Regulate Polarization of Macrophages

Therapeutic strategies reducing the M1 phenotype and promoting the M2 phenotype in the kidney macrophages have been attempted for DN [71]. Sodium-glucose cotransporter inhibitors (SGLT2is) have been recently proven to significantly affect renal outcomes in patients with DN via different mechanisms [72]. It has been shown that empagliflozin reduces M1-polarized macrophage accumulation while inducing the anti-inflammatory M2 phenotype of macrophages, lowering plasma TNFα levels, and attenuating obesity-related chronic inflammation [73]. Furthermore, dapagliflozin causes a shift from inflammatory M1 macrophages towards the M2-dominant macrophages and exerts direct anti-inflammatory effects by inhibiting the expression of TLR-4 and activation of NF-κB [74]. In addition, vitamin D suppresses macrophage infiltration by downregulating TREM-1 [75] and inhibiting macrophage transition into the M1 phenotype through the STAT-1/TREM-1 pathway in DN rats [76]. Rosiglitazone could ameliorate renal tubular injury that resulted from oxidative stress and the inflammatory response by suppressing M1 macrophage polarization and promoting M2 macrophage polarization [77]. Hyperoside (HPS) can improve the renal inflammatory response in mice by promoting the polarization of macrophages from the M1 to the M2 phenotype [78]. In addition, miRNAs from extracellular vesicles can also affect the polarization process of macrophages [79]. Since macrophage polarization is a dynamic process, and M2 macrophages may play a role in renal fibrosis, regulation of macrophage polarization may not be a good approach.

### 6.4. Increase Autophagy of Macrophages

It has been shown that high glucose reduces the activity of macrophage autophagy, which can promote macrophage adhesion and migration [80]. Activation of the autophagosome of macrophages could improve the autophagy flux, leading to a reduction in the macrophage’s adhesion and migration. Moreover, a study has demonstrated that mitophagy participates in the regulation of the M1/M2 macrophage phenotype in DN [81]. The loss of mitochondrial function resulted in ROS generation, which impaired lysosomes, and blocked autophagic flux in macrophages under diabetic conditions, while a defective autophagic flux further promoted macrophage polarization towards the M1 phenotype and produced inflammation [82]. Therefore, improving macrophage autophagy could be an attractive strategy to attenuate renal inflammation and injury in DN.

### 6.5. MSCs

The administration of mesenchymal stem cells (MSCs) has been shown to attenuate kidney injury in DN models. After infusion of umbilical cord-derived mesenchymal stem cells (UC-MSCs) in T2DM mice, glucose homeostasis, hepatic function, and dyslipidemia were greatly improved, and a significant increase in M2 macrophages was seen in a number of important organs, suggesting that UC-MSCs may regulate macrophage function [83]. Moreover, microRNA-146a-5p-modified human UC-MSCs can facilitate M2 macrophage polarization to protect the kidneys from injury [84]. Recent studies showed that MSCs communicated with macrophages in DN mice through exchanges in mitochondrial content to ameliorate kidney injury, and that this effect was mediated via PGC-1α-mediated mitochondrial biogenesis and PGC-1α/TFEB-mediated lysosome-autophagy [85]. It has been shown that TNF-α-induced gene/protein (TSG)-6 released by MSCs (mesenchymal stem cells) can reduce renal tubular cell injury by increasing the expression of M2 macrophage markers and reducing the adhesion and migration of M1 macrophages [86]. Therefore, MSCs could be another interesting approach to modify macrophage function in DN. However, the specificity of MSCs could be viewed as a concern.

### 6.6. Limitations

Targeting macrophages as a therapy for DN has several limitations. Systemic inhibition of macrophage infiltration may impair the tissue repairing process and fighting against bacterial infection in diabetic patients. Meanwhile, these studies do not likely represent physiological conditions, as it is rare for the whole organ to lose its tissue-resident macrophage population at once. In the presence of certain disturbances or stimuli, apoptotic macrophages can be partially replenished by tissue resident-derived and monocyte-derived macrophages [87]. Inhibition of the phagocytosis function of macrophages may also affect the immune defense function of macrophages. Therefore, it is important to develop kidney-specific drugs to target macrophage infiltration and activation by modifying the tissue microenvironment in diabetic kidneys. It is also important to better understand the dynamic role of macrophages in the progression of DN, and therefore we could target macrophages at a specific time.

## 7. Conclusions and Perspectives

Macrophages likely play a critical role in the pathogenesis of DN by inducing renal inflammation and fibrosis. Classification of macrophages into M1 and M2 helps us to better understand their roles in the regulation of inflammation and tissue repair process. However, macrophages have high plasticity and undergo phenotypic changes quickly in different microenvironments. Studies suggest that the diabetic kidney has both the increased infiltration of macrophages and the activation of resident macrophages. Both M1 and M2 macrophages increase in the diabetic kidney. Recent single-cell RNA sequencing studies suggest that macrophage polarization is a continuous process in the diabetic condition. Macrophages also interact closely with intrinsic renal cells. The phenotypes of macrophages are determined by the renal tissue microenvironment. Potential therapies targeting macrophages, such as the direct depletion of macrophages, modulation of the macrophages polarization, use of MSCs to increase M2 macrophages, increasing macrophage autophagy, and modulation of the renal tissue microenvironment are all worthy of further study. However, all these approaches have their limitations, and further studies are needed to develop better approaches by avoiding the limitations of these existing strategies.

## Figures and Tables

**Figure 1 biomedicines-11-01889-f001:**
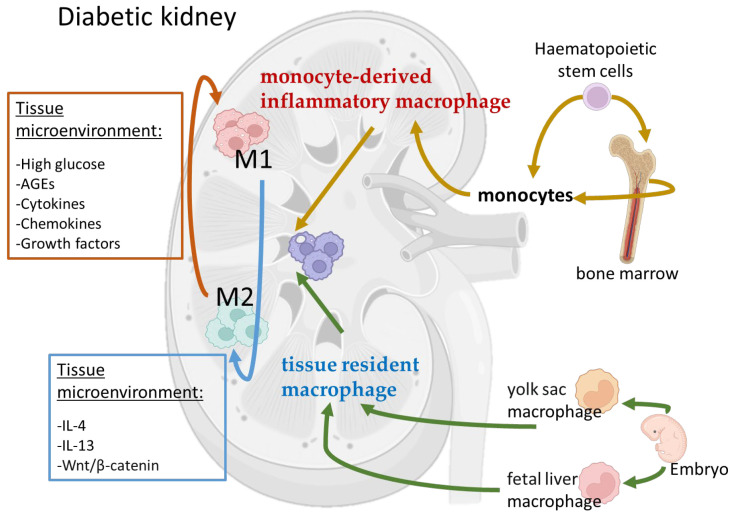
Macrophages have high plasticity to adapt in different microenvironments in diabetic kidneys. Kidney macrophages can be derived from the differentiation of blood monocytes or exist as resident cells in tissues from early embryonic development. After injury, kidney cells release cytokines and growth factors which, together with diabetic milieu, such as high glucose and AGEs, form a diseased microenvironment in the diabetic kidneys. This microenvironment induces monocyte infiltration and activation into inflammatory macrophages (M1). Tissue resident macrophages are also activated to form M1 macrophages. When the disease progresses to the late stage, the kidney microenvironment changes, and these macrophages change their phenotype from M1 and M2. However, this M1-to-M2 switch is a reversible and dynamic process.

**Figure 2 biomedicines-11-01889-f002:**
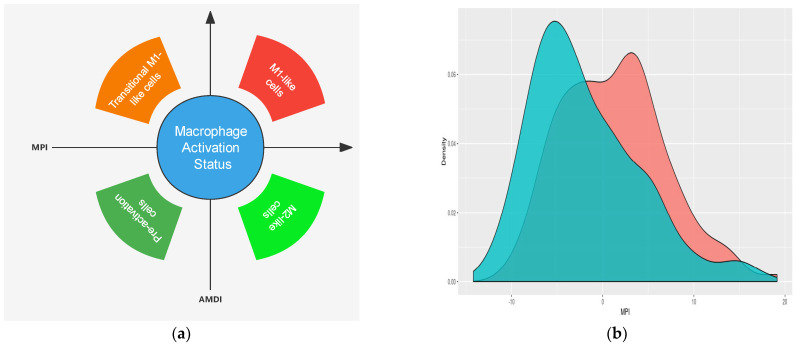
scRNAseq analysis revealing the dynamic transition in macrophage activation in the diabetic kidney. As described in our study [25], we utilized a single–cell transcriptome–based annotation tool MacSpectrum, which infers the macrophage activation status by estimating two indices: the macrophage polarization index (MPI) to annotate the degree of inflammation, and the activation–induced macrophage differentiation index (AMDI) to annotate the degree of terminal maturation (**a**). With this approach, the macrophages from control (WT) and diabetic kidneys from OVE26 mice (OVE) were mapped onto the MacSpectrum plot as “M1–like”, “M2–like”, “transitional”, and “pre–activation” phenotypes, and (**b**,**c**) show that macrophages from diabetic mice shifted to more inflammation (high MPI) and less differentiation (low AMDI). Consistent with this, (**d**) shows that macrophages shifted to the right (high MPI) and low (low AMDI) corner (Figure generated from the data published in the paper [25]).

**Table 1 biomedicines-11-01889-t001:** List of the studies on the interactions of macrophages with glomerular cells and tubular epithelial cells.

Ingredients	Targets	Results	References
M1 macrophages and podocytes	MCP-1	M1 macrophages increased podocyte permeability and damaged podocytes function; podocytes treated with high-glucose promoted macrophage migration and accumulation through secreting MCP-1	[31]
Macrophages and podocytes	TNF-α	Macrophages released TNF-α under high-glucose conditions and promoted the apoptosis of podocytes	[32]
Macrophages and podocytes	Tim-3, NF-κB/TNF-α	Macrophage activation induced by Tim-3 accelerated podocyte damage via the NF- κ B/TNF- α pathway	[33]
Macrophages EVs and podocytes	MiR-21-5p	MiR-21-5p in macrophage-derived EVs increased ROS production through the targeted inhibition of A20, leading to podocyte damage	[34]
Macrophages and podocytes	Vitamin D, FK506, Sirt6	Vitamin D, FK506, and overexpression of Sirt6 reduced podocyte damage by inhibiting the activation of M1 macrophages.	[35,36,37]
Macrophages and GECs	VEGF, NO, Flt-1	VEGF was chemotactic for macrophages, and negatively regulated VEGF-induced macrophage migration by inhibiting expression	[38]
Macrophages and GECs	ICAM-1, VCAM-1	Vascular endothelium-overexpressed cell adhesion molecules, including ICAM-1 and VCAM-1 homed circulating macrophages	[39,40]
Macrophages and GECs	GLP-1R	Exendin-4 (GLP-1R agonist) directly acted on the GLP-1R on GECs to reduce the expression of ICAM-1 and inhibit macrophage infiltration	[41]
M1 macrophages and GECs	ROS	The accumulation of M1 macrophages upregulated the ROS level in human GECs to promote cell damage	[42]
M1 macrophages and GECs	HIF-1α/Notch1, PPAR-α	Injured GECs upregulate the HIF-1α/Notch1 pathway in DN leading to M1 macrophage recruitment, which was reversed by the PPAR-α agonist fenofibrate to improve the GECs = function	[43]
Macrophages and RTECs	MCP-1, osteopontin	MCP-1 and osteopontin expressed by RTECs were critical factors, playing a key role in the communication between the injured RTECs and infiltrating macrophages under high-glucose conditions	[47,48]
Macrophages, glomeruli, and tubulointerstitium	TLR2	TLR2 was proven to be highly expressed in both the glomeruli and tubulointerstitium, and was associated with the increased renal expression of MyD88 and MCP-1, activation of NF-κB, and infiltration of macrophages	[49]
Macrophages and RTECs	TLR4	Increased expression of TLR4 in the renal tubules of human kidneys with DN correlated with interstitial macrophage infiltration as well as tubulointerstitial inflammation	[50]
Macrophages and RTECs	Necroptosis inhibitor necrostatin-1	Macrophages participated and promoted the necroptosis of RTECs in the high-glucose condition, which could be inhibited by the necroptosis inhibitor necrostatin-1	[28]
M1 macrophages and RTECs	IL-1β	High glucose stimulated IL-1β expression in RTECs to induce the M1 polarization of macrophages	[51]
M2 macrophages exosomes and podocytes	miR-25-3p	miR-25-3p in exosomes produced by M2 macrophages protected podocytes against HG-induced injury through activating autophagy in podocytes via inhibiting dual-specificity protein phosphatase 1 (DUSP1) expression	[52]
M2 macrophage exosomes and podocytes	miR-93-5p, TLR4	miR-93-5p expression was markedly upregulated in lipopolysaccharide (LPS)-induced podocytes, and inhibition of miR-93-5p or silencing of TLR4 reversed the reno-protective effects of miR-93-5p-containing exosomes produced by M2 macrophage on LPS-induced podocyte injury	[53]
RTECs EVs and macrophage	HIF-1α	EVs from HSA-treated RTECs can accelerate macrophage glycolysis by stabilizing HIF-1α expression	[55]
Urinary EVs and tubulointerstitial inflammation	miR-19b-3p	High levels of miR-19b-3p were found in urinary EVs and were correlated with the severity of tubulointerstitial inflammation in patients with DN	[56]
Urinary EVs and M1 macrophage	miR-199a-5p, Klotho/TLR4	Tail-vein injections of miR-199a-5p, which was found to be increased in the urinary EVs from diabetic patients with macroalbuminuria, induced kidney macrophage M1 polarization and accelerated the progression of DN by targeting the Klotho/TLR4 pathway	[57]
Macrophage EVs and RTECs	miR-7002-5p, Atg9b	miR-7002-5p in EVs derived from high glucose-induced macrophages suppressed autophagy of RTECs by targeting Atg9b, leading to renal tubular dysfunction and inflammation	[58]
RTECs exosomes and macrophage	Notch 1	Epsin1 modulated tubulointerstitial inflammation via the regulation of exosomal-Dll4 release from RTECs, which then activated Notch 1 signaling in macrophages under DN conditions	[59]
RTECs and macrophage	Macrophage–myofibroblast transition	Macrophages have been identified as the main source of myofibroblasts via macrophage–myofibroblast transition, thereby promoting renal fibrosis.	[60]
Macrophage exosomes and RTECs	LncRNAs	Exosomes from high glucose-treated macrophage promote RTECs to switch to a more pro-fibrosis phenotype via releasing long non-coding RNAs	[61]

## Data Availability

No new data were created or analyzed in this study. Data sharing is not applicable to this article.

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
