# Peer review of "Relationship between Macrophages and Tissue Microenvironments in Diabetic Kidneys"

_biomedicines, 2023, doi:10.3390/biomedicines11071889_

Round 1

Reviewer 1 Report

1- The abstract is lengthy, modify it, and explain some new therapies for DN in the conclusion part.

2- The keywords are not enough. At least 6 keywords are required. 

3- In section 5, first explain Glomerular cells,  Tubular epithelial cells, and Exosomes in brief. Then make a Table and compare the results of different studies regarding Glomerular cells,  Tubular epithelial cells, and Exosomes.

4- Section 6 is not described well. Each part needs to discuss deeply using different studies and conclude at the end of each part.

5- The following reference could be helpful to improve the manuscript: Sabbagh, F., Muhamad, I. I., Niazmand, R., Dikshit, P. K., & Kim, B. S. (2022). Recent progress in polymeric non-invasive insulin delivery. International Journal of Biological Macromolecules.

Minor editing of the English language required

Author Response

Thank you very much for your advice.  We have revised the paper one by one according to your comments. 

1- The abstract is lengthy, modify it, and explain some new therapies for DN in the conclusion part.

Response: We have modified the abstract according to the suggestions.

2- The keywords are not enough. At least 6 keywords are required. 

Response: We have added more keywords.

3- In section 5, first explain Glomerular cells, Tubular epithelial cells, and Exosomes in brief. Then make a Table and compare the results of different studies regarding Glomerular cells, Tubular epithelial cells, and Exosomes.

Response: We have provided the explanation accordingly and added a table for comparison.

4- Section 6 is not described well. Each part needs to discuss deeply using different studies and conclude at the end of each part.

Response: We revised this section according to the suggestions.

5- The following reference could be helpful to improve the manuscript: Sabbagh, F., Muhamad, I. I., Niazmand, R., Dikshit, P. K., & Kim, B. S. (2022). Recent progress in polymeric non-invasive insulin delivery. International Journal of Biological Macromolecules.

Response: We appreciate very much that you gave us many insightful feedbacks including this reference paper. We revised the manuscript based on your suggestions.

Comments on the Quality of English Language:Minor editing of the English language required

Response: We revised the language accordingly.

Reviewer 2 Report

REFERENCE NO 50 IS NOT CITED ANYWHERE IN MANUSCRIPT

Author Response

Thank you very much for your feedback. We apologize that we did not update the EndNote correctly, which resulted in such an error. Errors have been corrected and the references numbers are now all consecutive in the manuscript.

Reviewer 3 Report

I read with great interest “Relationship between Macrophages and Tissue Microenvironments in Diabetic Kidneys” by Yan et al.

Paper design is fine. The article is logically divided into sections and subsections. English is fine, only minor spell check.

Comments:

1.      I would provide more details on the macrophage’s recruitment in the kidney tissue. This is also crucial to highlight an important association between inflammation and the vicious circle it will create. In fact, more inflammation more macrophages, more kidney damage and so on.

2.      The role of oxidative stress is reported in the abstract, but not in the text. Please improve.

3.      It would be nice to also report how macrophages changes according to antidiabetic therapies. In particular SGLT2is, have proven to ameliorate renal function and to reduce inflammation in several studies and trials (doi: 10.31083/j.rcm2303106)

English is fine, only minor spell check required.

Author Response

Thank you for your feedback. We have modified the manuscript according to your comments.

  1. I would provide more details on the macrophage’s recruitment in the kidney tissue. This is also crucial to highlight an important association between inflammation and the vicious circle it will create. In fact, more inflammation more macrophages, more kidney damage and so on.

Response: We have revised the paper accordingly.

  1. The role of oxidative stress is reported in the abstract, but not in the text. Please improve.

Response: We eliminated oxidative stress in both abstract and text to be consistent.

  1. It would be nice to also report how macrophages changes according to antidiabetic therapies. In particular SGLT2is, have proven to ameliorate renal function and to reduce inflammation in several studies and trials (doi: 10.31083/j.rcm2303106)

Response: We have added this in the discussion according to the suggestions.

Comments on the Quality of English Language: English is fine, only minor spell check required.

Response: We have checked spelling carefully.

Reviewer 4 Report

The manuscript of “Relationship between Macrophages and Tissue Microenvironments in Diabetic Kidneys” by Jiayi Yan and co-authors aims to review the interactions between macrophages, intrinsic renal cells, and tissue microenvironment in the diabetic kidney. The authors summarized recent knowledge about the dynamic role of macrophages in regulation of inflammation and tissue repair process in diabetic kidneys. The strong point of the review is the analysis of single cell RNA-sequencing data, which significantly advanced our understanding of the classification of macrophages based on their unique molecular markers. The authors concluded that it is important to develop kidney-specific drugs to target macrophage infiltration and activation by modifying tissue microenvironment in diabetic kidneys.

The manuscript is quite interesting and well written; all the conclusions are supported by the data obtained. The topic of the manuscript is highly relevant and timely in view of recent the statistics on the incidence of diabetes kidneys in the world. The manuscript contributes to the systematization of current knowledge about the involvement of the reversible and dynamic process of switching between M1 and M2 macrophage phenotypes in the kidneys in diabetic conditions. The manuscript may be accepted for publication after minor revision.

Comments:

1.     The review lacks some generalizing illustrations or tables that would show in detail the differences between M1 and M2 macrophage phenotypes and/or factors and conditions that contribute to switching between these phenotypes.

2.     In the Conclusions section, the most promising therapeutic strategies for diabetic kidneys remained unrevealed. The section could be improved by explaining whether the therapeutic strategies by reducing the M1 phenotype and promoting the M2 phenotype in kidney macrophages may be promising.

Author Response

Your comments are greatly appreciated. We hvae revised the manuscript based on your suggestion.

  1. The review lacks some generalizing illustrations or tables that would show in detail the differences between M1 and M2 macrophage phenotypes and/or factors and conditions that contribute to switching between these phenotypes.

Response: We have updated figure 1 and figure 2 and added a new table to illustrate this.

  1. In the Conclusions section, the most promising therapeutic strategies for diabetic kidneys remained unrevealed. The section could be improved by explaining whether the therapeutic strategies by reducing the M1 phenotype and promoting the M2 phenotype in kidney macrophages may be promising.

Response: We have added this in the conclusion section.

Reviewer 5 Report

Macrophages play a significant role in the development and progression of diabetic nephropathy (DN), which is a common complication of diabetes that affects the kidneys. In this review, Yan et al elaborated the mechanism of macrophages on the development of DN, such as the M1 and M2 macrophage classification, the M1/M2 ratio in human DN, and Potential therapeutics for DN by targeting macrophages. This is a well done, in dept study. There are only a few points that should be addressed.

1.     We typically refer to the classification of macrophages as M1 and M2 as macrophage polarization. It is suggested that the author change the title "Classification of Macrophages in DN" to "Macrophage Polarization in DN."

2.     The M1/M2 ratio is very important to the development of diseases. I suggest the authors introduce more about the M1/M2 ratio and development of DN.

3.     Macrophages are innate immune cells. In my opinion, the introducing the origin of this type of cell would be highly meaningful. For instance, there is a report published on Nat Rev Immunol this year about the tissue-specific macrophages (doi: 10.1038/s41577-023-00848-y), and there are also papers related to tissue-resident macrophages in kidney, such as doi: 10.3389/fphys.2017.00837. I suggested the authors add some of the information about these.

4.     In my opinion, the references of the author are somewhat outdated. I recommend the author to increase the proportion of papers from the past five years.

Author Response

We really appreciate your advice. We revised the manuscript according to your suggestion.

  1. We typically refer to the classification of macrophages as M1 and M2 as macrophage polarization. It is suggested that the author change the title "Classification of Macrophages in DN" to "Macrophage Polarization in DN."

Response: We have changed the title according to your suggestion.

  1. The M1/M2 ratio is very important to the development of diseases. I suggest the authors introduce more about the M1/M2 ratio and development of DN.

Response: We discussed this in human with DN.

  1. Macrophages are innate immune cells. In my opinion, the introducing the origin of this type of cell would be highly meaningful. For instance, there is a report published on Nat Rev Immunol this year about the tissue-specific macrophages (doi: 10.1038/s41577-023-00848-y), and there are also papers related to tissue-resident macrophages in kidney, such as doi: 10.3389/fphys.2017.00837. I suggested the authors add some of the information about these.

Response: We have added this discussion and the above reference in the text.

  1. In my opinion, the references of the author are somewhat outdated. I recommend the author to increase the proportion of papers from the past five years.

Response:  We have updated some of the references.

Round 2

Reviewer 3 Report

The author appropriately answered to all the issues I raised.

Reviewer 5 Report

The author has solved all my problems and I think it can be published.